# Reducing Liver Cancer Risk through Dietary Change: Positive Results from a Community-Based Educational Initiative in Three Racial/Ethnic Groups

**DOI:** 10.3390/nu14224878

**Published:** 2022-11-18

**Authors:** Lin Zhu, Ellen Jaeseon Kim, Evelyn González, Marilyn A. Fraser, Steven Zhu, Nathaly Rubio-Torio, Grace X. Ma, Ming-Chin Yeh, Yin Tan

**Affiliations:** 1Center for Asian Health, Lewis Katz School of Medicine, Temple University, Philadelphia, PA 19140, USA; 2Department of Urban Health and Population Science, Lewis Katz School of Medicine, Temple University, Philadelphia, PA 19140, USA; 3Office of Community Outreach, Fox Chase Cancer Center, Temple University Health System, Philadelphia, PA 19111, USA; 4Arthur Ashe Institute for Urban Health, New York, NY 11203, USA; 5Pennsylvania United Chinese Coalition, Philadelphia, PA 19107, USA; 6Nutrition Program, Hunter College, City University of New York, New York, NY 10017, USA

**Keywords:** health promotion, liver cancer prevention, community-based participatory research, dietary behaviors, racial/ethnic minority, academic-community partnership

## Abstract

Dietary behaviors and alcohol consumption have been linked to liver disease and liver cancer. So far, most of the liver cancer awareness campaigns and behavioral interventions have focused on preventive behaviors such as screening and vaccination uptake, while few incorporated dietary aspects of liver cancer prevention. We implemented a community-based education initiative for liver cancer prevention among the African, Asian, and Hispanic populations within the Greater Philadelphia and metropolitan New York City areas. Data from the baseline and the 6-month follow-up surveys were used for the assessment of changes in dietary behaviors and alcohol consumption among participants. In total, we recruited 578 participants through community-/faith-based organizations to participate in the educational workshops. The study sample included 344 participants who completed both baseline and follow-up survey. The Hispanic subgroup was the only one that saw an overall significant change in dietary behaviors, with the Mediterranean dietary score increasing significantly from 30.000 at baseline survey to 31.187 at 6-month follow-up assessment (*p* < 0.05), indicating a trend towards healthier dietary habit. In the African Americans participants, the consumption scores of fruits and poultry increased significantly, while vegetables and red meats decreased. In Asian Americans, the consumption of non-refined cereals, red meats, and dairy products decreased. Alcohol consumption decreased significantly among Hispanics while it did not change significantly among the other two communities. This community-based educational initiative generated different impacts in the three populations, further highlighting the needs for more targeted, culturally tailored efforts in health promotion among these underprivileged communities.

## 1. Introduction

Liver cancer is one of the fastest-growing cancer-cause of death in the United States [1,2]. According to the American Cancer Society, the incidence of liver cancer has tripled since the 1980s and the death rates more than tripled in the US [3]. Liver cancer incidence and mortality rates in African Americans, Asian Americans and Hispanic Americans were significantly higher than that among non-Hispanic whites [4,5,6,7,8]. Hepatitis B virus (HBV) and hepatitis C virus (HCV) infection contributes to 65% of hepatocellular carcinoma (HCC) in the US, the most common subtype of primary liver cancer [9,10,11,12,13]. Research estimated that about 39%, 47%, and 22% of HCC cases were attributable to HBV/HCV infection in African Americans, Asian Americans, and Hispanics, respectively [13]. Particularly noteworthy was the significant role of metabolic disorders. Between 2000 and 2011, the population attributable fraction of metabolic disorders to HCC increased from 26% to 36% [13]. Metabolic disorders accounted for 32% of HCC cases among Hispanics, the highest among all racial/ethnic groups examined [13]. Given the increase in metabolic disorder prevalence in multiple populations, it was possible that this was a driving force in the increases in HCC rates among the Hispanic population. 

In 2016, the World Health Assembly adopted the Global Health Sector Strategy (GHSS), which has called for the global elimination of viral hepatitis B and C infection as a public health problem and 65% reduction in mortality by 2030 [14]. The Viral Hepatitis National Strategic Plan for the United States has also urged for the expansion of culturally competent and linguistically appropriate viral hepatitis prevention services [15]. Despite growing public health efforts in viral hepatitis and liver cancer prevention, vaccination, and treatment services, the burden of HBV and HCV is still much heavier in racial/ethnic minorities [15,16,17,18]. Structural barriers to healthcare services, systemic racism and discrimination, as well as inequities in society and individual level social determinants of health contributed to the considerable disparities in viral hepatitis infection, exposure, immunity, and mortality [6,19,20,21,22]. 

In the past three decades, there has been growing efforts to implement culturally tailored programs that aimed to raise knowledge and awareness of viral hepatitis, promote HBV/HCV screening, and increase HBV vaccine uptake or HCV treatment adherence [23,24,25,26,27]. However, few programs have incorporated lifestyle components in the context of viral hepatitis and liver cancer prevention. An increasing body of literature has suggested the modifiable lifestyle behaviors such as physical activity, alcohol consumption, and dietary behaviors can impact risks of liver cancer and liver related diseases [28,29,30,31,32]. Specifically, higher intake of whole grain and dietary fiber have been linked to lower risks of liver cancer and liver disease mortality [32]. In addition, food groups such as red and processed meats that include saturated fats, as well as high sodium consumption, were found to be positively correlated with HCC occurrence whereas high consumption of white meat, fish, fiber, vegetables, and fruits were associated with lower HCC risks [33,34,35,36,37]. Alcohol has been a confirmed risk factor of liver cancer [29]. It was estimated that one alcoholic drink (about 12 grams) might be associated with a 1.1 times higher risk for liver cancer. Lifestyle interventions have shown to be effective in promoting healthy behaviors and health service utilization [38,39,40]. Promoting healthy behavioral changes by utilizing and reinforcing perceived self-efficacy, subjective norms, enabling factors, and attitudinal changes has shown to be crucial in helping participants adopt and adhere to a healthy lifestyle in the contexts of disease prevention or chronic illness management [38]. However, such holistic wellness perspectives have yet to be widely adopted in liver cancer prevention programs. 

The goal of this study was to assess the changes in dietary and alcohol consumption behaviors achieved through a culturally tailored, community-based educational initiative in African American, Asian American, and Hispanic/Latinx communities in New York City and the greater Philadelphia metropolitan area. 

## 2. Materials and Methods

### 2.1. Study Design and Participants

The Synergistic Partnership for Enhancing Equity in Cancer Health (SPEECH) is a comprehensive regional cancer health disparity partnership between Temple University Fox Chase Cancer Center and Hunter College (TUFCCC/HC), the U54 grant funded by the National Cancer Institute (grant number: U54 CA221704). The purpose of SPEECH is to reduce cancer health disparities among underserved minority populations in Pennsylvania-New Jersey-New York City (PNN) region, through cancer disparities research, community outreach, and career development for underrepresented early-stage investigators and students. One major component of the SPEECH Partnership is the Community Outreach Core (COC). The overall goal of the Community Outreach Core is two-fold: (1) strengthen community engagement in cancer outreach research through evidence-based community cancer education initiatives and outreach activities to reduce cancer disparities among underserved African Americans, Asian Pacific Americans and Hispanic populations in the PNN region, and (2) provide a robust and sustainable community-based participatory research infrastructure for the TUFCCC/HC Partnership.

Building on previously established relationships with community-based organizations (CBOs) and through outreach to new organizations, the COC team engaged 50 CBOs serving the African American, Asian American, and Hispanic/Latinx communities in New York City and the greater Philadelphia metropolitan area from 2018 to 2021. The CBO leaders and staff were actively engagement developing participant recruitment strategies, conducting outreach, and in delivering the educational workshops and conducted data collection through surveys. To be eligible to participate in this initiative, participants must be at least 18 years old to participate in this initiative. In total, we were able to recruit 578 participants to participate in the educational initiative, specifically, the education workshops co-delivered by trained bilingual community health workers affiliated with collaborating CBOs. It was worth noting that with support of the TUFCCC/HC Partnership, we were able to successfully transition community workshops from in-person to the virtual space during the COVID-19 pandemic period. 

### 2.2. Data Collection

To assess the impact of this educational workshop on community members’ knowledge and awareness of viral hepatitis/liver cancer prevention, as well as their preventive behaviors, we collected data through face-to-face, telephone, or online survey at three time points: pre-workshop, post-workshop, and a 6-month follow-up survey. In total, 578 participants completed the pre-workshop survey, and 541 completed the post-workshop survey. Largely due to the disruptions brough by the COVID-19 pandemic, we were only able to reach 344 participants in the 6-month follow-up survey. 

### 2.3. Community Educational Curriculum

We adapted several educational curricula that had proven to be effective in raising cancer prevention awareness and behavior in minority populations in our previous studies [23,41,42] for this educational initiative for three minority populations. We incorporated facts and guidelines on liver cancer and viral hepatitis symptoms, prevention, and management from the National Cancer Institute, (cancer.gov), the Center for Disease Control and Prevention (CDC.gov), the Mayo Clinic (MayoClinic.org), and the American Cancer Society (Cancer.org). The curriculum included four modules: (1) unique liver cancer disparity in African/Black, Asian, and Hispanic population, (2) risk factors of liver cancer, (3) knowledge about HBV/HCV, (4) healthy eating and liver cancer prevention.

The Community Advisory Board (CAB) consisting of leaders representing each population and geographic area, hepatologists and gastroenterologists serving the minority populations of interests, as well as patient advocates were actively engaged in the design of not only the contents of the educational curriculum but also the format of education workshop delivery. Furthermore, we conducted a plain language evaluation on the education contents with the Health Literacy Advisor™ (in Spanish—Asesor de Comunicación en Salud™). This interactive software calculates readability statistics and identifies complex terminology, in addition to providing suggestions for replacement terms. In addition, we used the Flesch-Kincaid Readability Test Tool to evaluate the reading level of the informed consent forms to identify the pre-revision reading level of the document. We made subsequent adjustments to ensure that the contents were accessible and appropriate to our target audience. The final educational materials and data collection tools were at 6th to 8th grade reading level. Because the target populations included a large proportion of individuals with limited English proficiency, we translated all contents into Spanish and mandarin Chinese. Translations were reviewed by CAB members, healthcare providers, and community member representatives. 

CAB members and community representatives played a key role in tailoring the facts, guidelines, and recommendations on dietary and nutritional guidelines in relations to liver health, disease prevention, and management. For example, in explaining what a serving of fruit or vegetables look like, CAB members representing the Asian communities recommended that we use examples of fruits and vegetables that were more commonly consumed in the Asian communities, such as boy choy as a typical vegetable. Similarly for the other food categories, we tailored the content to include examples and recommendations based on the existing patterns of food consumptions of the racial/ethnic group. For example, for the Asian communities, we added tofu as a good source for protein since tofu was readily available and widely incorporated in lots of Asian cuisine. For African American and Hispanic communities, we also consulted with CAB members from the communities to make sure that the examples given were appropriate and accessible to the target audience. Such efforts ensured that the health promotion messages on dietary habits were not only apprehensive to the target audience, but also appropriate and practical. 

### 2.4. Measurement 

We assessed participants’ dietary behaviors with the Mediterranean dietary score [43]. This measurement included 11 specific behaviors, including the consumptions of 10 food groups and alcohol intake. The food groups included non-refined cereals (whole bread, pasta, rice, other grain, biscuits, etc.), fruit, vegetables, legumes, potatoes, fish, meat and meat products, poultry, full fat dairy products (like cheese, yoghurt, and milk), as well as olive oil. Monotonic functions were used to score the frequency consumption of these foods. For the consumption of items that were considered healthy, we assigned score 0 when participants reported no consumption, 1 for 1–4 servings per month, 2 for 5–8 servings per month, 3 for 9–12 servings per month, 4 for 13–18 servings per month, and 5 for more than 18 servings per month. These groups included non-refined cereals, fruits, vegetables, legumes, fish, potatoes, and olive oil. For the consumption of items that were considered not as healthy (including meat and meat products, poultry and full fat dairy products), we first calculated the total amount of ethanol concentration from the reported consumption of different kinds of alcohol (wine, beer, and liquor/spirit). We then assigned the scores on a reverse scale. For alcohol, we assigned score 5 for consumption of less than 300 mL of alcohol per day, 4 for 300–399 mL per day, 3 for 400–499 mL per day, 2 for 500–599 mL per day, 1 for 600–699 mL per day, and 0 for 700 mL or more per day. The total Mediterranean score ranged from 0 to 55 with a higher numeric value indicating a greater adherence to the Mediterranean diet, i.e., a healthier diet. 

### 2.5. Statistical Analysis 

All analyses were conducted in R [44]. We used chi-square tests to compare the categorical demographic characteristics and dietary behaviors across the three racial/ethnic groups, and ANOVA for continuous characteristics and behaviors. We conducted repeated-measure ANOVA to examine the changes of the specific individual food group dietary scores and the total Mediterranean diet score from baseline survey to 6-month follow-up survey for each of the three racial/ethnic groups. We used the Bonferroni correction to adjust the *p*-value reported in the repeated-measure ANOVA results (Figure 1). We also fitted a linear mixed-effects model to examine the change of Mediterranean diet score while controlling for sex, gender, education level, and health insurance coverage for each racial/ethnic subgroup. Additionally, we conducted Tukey’s honestly significant difference test to conduct multiple comparisons and assess the significance of differences between the baseline and 6-month follow-up Mediterranean diet score. Two-sided *p* values < 0.05 were considered statistically significant. 

## 3. Results

### 3.1. Demographic Characteristics

The study sample consists of 344 participants who completed both baseline and 6-month follow-up survey. Among them, 54% identify as Asian American, 15% as African American, and 30% as Hispanic American. The majority of the overall sample was female (71.5%), born outside of the US (81.5%), did not have a college degree (72.5%), and did have health insurance (80.6%). The three racial/ethnic groups differed on all sociodemographic characteristics except for gender composition (Table 1). 

### 3.2. Dietary Changes 

We compared the scores of 11 specific food categories and the overall Mediterranean diet scores between baseline and the 6-month follow-up assessment within each of the three racial/ethnic groups (Table 2). Among African American participants, the consumption scores of fruits (2.170 to 3.581, *p* < 0.05) and poultry (2.613 to 3.677, *p* < 0.01) significantly increased, while the consumption of vegetables (3.419 to 2.677, *p* < 0.05 and red meat (3.645 to 2.065, *p* < 0.001) decreased significantly. We did not see significant changes in other dietary categories. In Asian Americans, the scores of non-refined cereals (3.362 to 4.282, *p* < 0.001), fruits (3.385 to 4.374, *p* < 0.001), red meat (2.655, 3.195, *p* < 0.001), poultry (2.420 to 3.736, *p* < 0.001), and dairy products (2.851 to 3.701, *p* < 0.001) significantly increased; consumption of vegetables (3.908 to 3.236, *p* < 0.001) decreased from baseline to 6-month follow-up assessment. In Hispanic participants, the consumptions of poultry (2.173 to 3.554, *p* < 0.001), dairy products (2.101 to 3.863, *p* < 0.001), olive oil (3.022 to 3.489, *p* < 0.01) went up significantly while consumptions of vegetables (3.079 to 2.849, *p* < 0.001) and red meat (2.849 to 2.532, *p* < 0.05) decreased. In addition, the Hispanic sample was the only group that saw changes in alcohol consumption, with the score increasing from 4.360 to 4.688 (*p* < 0.05), indicating a decrease in alcohol consumption. It was also the only group that saw a significant changes in the total Mediterranean diet score, specifically, a slight increase from 30.000 at baseline survey to 31.187 at 6-month follow-up assessment (*p* < 0.05), indicating a trend towards healthier dietary habit on average among this sub-sample (Figure 1).

We fitted a linear mixed-effects model to examine the change of Mediterranean diet score while controlling for sex, gender, education level, and health insurance coverage. The results (Table 3) showed that the effects of time on the outcome variable was significant (*p* < 0.01), with the sociodemographic variables accounted for in the model. Additionally, we conducted Tukey’s honestly significant difference test to assess the significance of differences between the baseline and 6-month follow-up Mediterranean diet score, using the *multcomp* package in R, which offers a convenient interface to perform multiple comparisons in a general context [45]. The results indicated that the increase of the Mediterranean diet score from baseline to 6-month follow-up assessment was statistically significant. Changes in the Mediterranean diet score were not statistically significant in Asian American and African American subsamples.

## 4. Discussion

We designed and implemented a culturally tailored educational initiative on liver cancer and viral hepatitis prevention among three racial/ethnic groups in two major metropolitan areas on the east coast of the United States. The findings showed that changes in dietary behaviors were mostly positive, but such changes varied by food categories and by racial/ethnic groups. It was important to note that the changes must be interpreted in the context of the COVID-19 pandemic. There has been a growing body of literature that examined how the COVID-19 pandemic has impacted the dietary habits in the US population. One consistent finding from the existing literature is that the COVID-19 pandemic has seen changes in both healthier and unhealthier trends in people’s eating habits [46,47,48,49,50,51,52]. People’s access to fresh produce which was impacted by lockdown measures and supply chain issues, COVID-19 related income loss, increased stress and anxiety, reduced outdoors time, increased exposure to food advertisement on the internet or television, and other factors caused by the COVID-19 pandemic further exacerbated the disparities in nutrition awareness, access to fresh produces, and food security, particularly for racial/ethnic minority populations [46,47,48]. All these factors, plus the heightened health awareness due to the pandemic, were the contexts in which our educational initiative took place in the target populations. 

Our findings were consistent with the mixed changes of dietary habits found in previous studies [46,47,48,49,50,51,52]. Specifically, we found that all three groups increase in both healthy and unhealthy dietary behaviors. For example, the African American participants increased their consumptions of fruits (healthy) but also increased red meat consumption (unhealthy); the Asian American sample were consuming more non-refined cereals and poultry (healthy) but also more red meat and dairy products (unhealthy); the Hispanic sample cut back on red meat consumption (healthy) but increased dairy and alcohol consumption (unhealthy). The Hispanic sample was the only population that showed an overall positive change in dietary behaviors, with their Mediterranean diet score significantly increasing from baseline to 6-month follow-up assessment. Additionally, this increase was significant even when we controlled for age, sex, education, and health insurance coverage. Possible reason for changes found in Hispanic sample might be contributed by program curriculum’s emphasis about sharped increased HCC in this population heavily associated with metabolic and dietary factors. These findings suggested that our culturally tailored education workshops, despite the severe disruptions the COVID-19 pandemic brough to people’s daily life, was able to generate some moderately positive impact on our target populations, especially in the Hispanic/Latinx sub-sample, which saw a slight yet statistically significant increase in the Mediterranean diet score. 

In the context of liver cancer and viral hepatitis prevention, the improvement in healthy dietary behaviors in the Hispanic/Latinx community, even the lack of significant changes in the other two communities, could indicate a positive impact of our educational initiative on the target populations. Community-based, culturally tailored educational programs have proven to be effective in promoting health dietary behaviors, especially for racial/ethnic minority communities [53,54,55]. By delivering the educational workshops in Chinese and Spanish, and in collaboration with the CBO leaders and staff members, we were able to remove the language barriers that our participants had previously experienced in accessing liver cancer and viral hepatitis prevention information. We also promptly responded to the COVID-19 outbreak by including useful information on community and healthcare resources for the participants in our educational workshops. 

Consumptions of potatoes, legumes, and fish did not change significantly in any of the racial/ethnic groups from baseline to 6-month follow-up. It was not immediately clear why intakes of these food categories remained stable. It could be that people’s access to these food categories were particularly limited due to store closure or supply chain issues. It was also possible that these food categories made up only a small portion of the overall diet, which meant smaller space for changes. Since legumes and fish both served as important sources of protein [56,57], and legumes also provided nutritionally important dietary fiber [57,58], future research should better understand how to better promote healthy dietary habits with regard to these food categories. In addition, the alcohol consumption did not change significantly for African American and Asian American subsamples, while it significantly decreased for the Hispanic subsample. In contrast, studies have found increased alcohol consumption during the COVID-19 pandemic. For example, data on a nationally representative sample of 1540 adults showed significant increase in alcohol consumption, especially among women [59]. Another study of 993 participants aged 21 and older also found increased alcohol consumption in general, and higher increases in excessive drinking rate among women and non-Hispanic black subgroup [60]. The fact that participants of our initiative did not significant increase their alcohol consumption spoke to the positive impact the culturally tailored education workshops had on lifestyle behaviors of the participants.

This educational initiative’s impact was not limited to the changes in participants’ perception and behaviors of cancer prevention. The CBPR-based strategies in CBO and participant recruitment, educational curriculum development, delivery, and evaluation, on top of being key to the success of this educational initiative, were instrumental in supporting subsequent health promotion initiatives and programs that the COC has been implementing. Even after the end of the recruitment process, several of our participating sites reached out to ask for more education sessions. Other CBOs reached out and expressed strong interests in participating in the current and/or future initiatives on cancer or chronic illness prevention. The relationships developed, mutual trust and respect established, in this initiative not only ensured the success of this program but will also provide an extraordinary foundation for the other community engagement activities and cancer health disparities research under the SPEECH partnership. 

In addition, feedback from the participants to our research team indicated that the results achieved in their participation in this initiative was attributed to the trust, leadership, and communication skills of the multilingual team of community health educators, coordinators, and CBO staff members. Together they were able to speak to the community members in languages that were respectful, appropriate, and accessible. Furthermore, receiving training on cancer prevention and delivering the educational workshops with our team helped the community health educators and CBO staff members strengthen their abilities in community outreach and health promotion. This is an epitome of community capacity building component of this initiative, a testimony of the impact that the SPEECH Partnership has been generating in the communities we serve.

Our study has a few limitations. Firstly, our participants are concentrated in the greater Philadelphia metropolitan area and the New York City area; thus, our findings may not be generalizable to the national population of these three racial/ethnic groups. Secondly, the dietary behaviors were self-reported, which might be affected by reporting errors or biases. Thirdly, it was impossible for us to disentangle the impact of our educational initiative from the impact of the COVID-19 pandemic. Whether the Mediterranean diet score was an appropriate measure of dietary behaviors for the racially diverse populations is another potential issue. The Mediterranean diet score, in a few slightly modified forms, has been widely used to examine dietary behaviors and profiles in diverse populations, including Asian American [61], African American [62,63,64,65], and Hispanic/Latinx [61,63,65,66] populations in the US, as well as populations in Asian countries [67,68,69]. Therefore, we consider it to be a valid measure for our study sample.

In addition, we note the suboptimal retention rate (64% from post-education survey to 6-month follow-up assessment). This was largely due to the disruptions to employment, education, and daily routines that the COVID-19 brought to the target populations. The retention rate was particularly low among the African American communities (26%), mostly located in the hardest hit neighborhoods in the greater Philadelphia metropolitan area [70] and the greater NYC metropolitan area [71]. It was noteworthy that this community initiative was able to build a robust and sustainable academic-community partnership through mutual respect and trust, shared power, as well as acknowledge, visibility, and recognition. Through efforts on community capacity and infrastructure building, we established the channels for timely communication and efficient logistical support between the academic institutions and the underserved communities. Since the outbreak of COVID-19 in March 2020, this initiative has incorporated informational and logistic support to the communities, disseminating accurate medical and public health information on COVID-19, providing logistic support and patient navigation to community members on healthcare access, and even organizing a COVID-19 vaccine mobile unit to low-income communities in Northeast Philadelphia. Responding to the actual needs in the communities in a timely fashion is key in maintaining a sustainable partnership with the communities and strengthening community resilience and empowerments.

In summary, the findings of this study contributed significantly to the growing body of literature on liver cancer and viral hepatitis prevention, dietary behavioral changes, and racial/ethnic disparities. The nuanced disparities in dietary habits in the three racial/ethnic minorities, and how they changed over time, provided meaningful epidemiological data for future studies on dietary patterns and nutrition in these populations, and future interventions on chronic disease prevention through promoting healthy dietary habits. 

## Figures and Tables

**Figure 1 nutrients-14-04878-f001:**
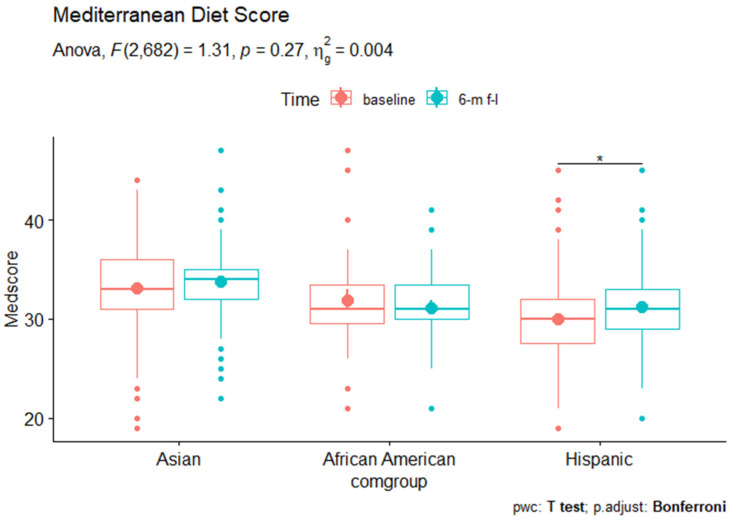
The Mediterranean diet scores among Asian American, African American, and Hispanic subgroups at baseline vs. 6 m follow up. * *p* < 0.05.

**Table 1 nutrients-14-04878-t001:** Demographic characteristics by racial/ethnic groups.

	African Americans (N = 31)	Asian Americans (N = 174)	Hispanic (N = 139)	Total (N = 344)	*p*-Value
Gender					ns
male	6 (6.1%)	47 (48.0%)	45 (45.9%)	98 (28.5%)	
female	25 (10.2%)	127 (51.6%)	94 (38.2%)	246 (71.5%)	
Age					<0.001
Mean (SD)	59.828 (18.989)	64.081 (12.210)	54.626 (16.743)	59.865 (15.465)	
Nativity status					<0.001
foreign-born	14 (5.1%)	168 (61.3%)	92 (33.6%)	274 (81.5%)	
US-born	16 (25.8%)	4 (6.5%)	42 (67.7%)	62 (18.5176/%)	
Marital status					<0.001
currently married	6 (3.4%)	126 (71.6%)	44 (25.0%)	176 (53.3%)	
other	24 (15.6%)	44 (28.6%)	86 (55.8%)	154 (46.7%)	
Education level					<0.001
<college	14 (5.6%)	135 (54.4%)	99 (39.9%)	248 (72.5%)	
college or above	17 (18.1%)	39 (41.5%)	38 (40.4%)	94 (27.5%)	
Health insurance					0.017
no	3 (4.8%)	25 (39.7%)	35 (55.6%)	63 (19.4%)	
yes	27 (10.2%)	142 (53.4%)	97 (36.5%)	266 (80.6%)	

**Table 2 nutrients-14-04878-t002:** Comparison of dietary scores among three groups at baseline vs. 6 m follow up.

Scores	African Americans			Asian Americans			Hispanics		
*Mean (sd)*	Baseline	6 Months	*p*-Value	Baseline	6 Months	*p*-Value	Baseline	6 Months	*p*-Value
Non-refined cereals	2.161 (1.416)	2.387 (1.585)	ns	3.362 (1.555)	4.282 (1.029)	<0.001	2.662 (1.432)	2.446 (1.533)	ns
Potato	1.258 (0.855)	1.097 (0.700)	ns	1.902 (1.181)	1.937 (1.128)	ns	1.957 (1.166)	1.734 (1.033)	ns
Fruits/100% Juice	2.710 (1.419)	3.581 (1.501)	<0.05	3.385 (1.504)	4.374 (1.022)	<0.001	30.058 (1.278)	3.252 (1.415)	ns
Vegetables	3.419 (1.409)	2.677 (1.400)	<0.05	3.908 (1.309)	3.236 (1.512)	<0.001	3.079 (1.246)	2.403 (1.540)	<0.001
Legume	2.419 (1.523)	2.323 (1.579)	ns	2.305 (1.323)	2.379 (1.180)	ns	2.748 (1.325)	2.885 (1.346)	ns
Fish	2.742 (1.365)	2.065 (1.340)	ns	2.977 (1.351)	2.718 (1.136)	ns	2.014 (1.291)	1.914 (1.182)	ns
Red meat	3.645 (1.018)	2.065 (0.892)	<0.001	2.655 (1.252)	3.195 (1.090)	<0.001	2.849 (1.268)	2.532 (1.229)	<0.05
Poultry	2.613 (1.174)	3.677 (1.447)	<0.01	2.420 (1.399)	3.736 (1.235)	<0.001	2.173 (1.313)	3.554 (1.303)	<0.001
Dairy	2.742 (1.483)	3.129 (1.839)	ns	2.851 (1.467)	3.701 (1.854)	<0.001	2.101 (1.264)	3.863 (1.441)	<0.001
Olive oil	3.097 (1.795)	3.097 (1.469)	ns	2.488 (1.712)	3.098 (1.749)	<0.005	3.022 (1.496)	3.489 (1.343)	<0.01
Alcohol	4.806 (0.910)	4.696 (1.063)	ns	4.747 (1.023)	4.787 (0.875)	ns	4.360 (1.460)	4.688 (1.103)	<0.05
Total Score	31.935 (5.785)	31.161 (4.140)	ns	33.080 (4.640)	33.782 (3.734)	ns	30.000 (4.535)	31.187 (4.124)	<0.05

Abbreviations: sd = standard deviation; ns = not significant.

**Table 3 nutrients-14-04878-t003:** Fixed effect estimates of the linear Mixed-effects model analysis of the Mediterranean diet score among the Hispanic subsample (N = 139).

	Asian American		African American		Hispanic	
Predictors	Estimates (Standard Error)	*p*-Value	Estimates (Standard Error)	*p*-Value	Estimates (Standard Error)	*p*-Value
(intercept)	27.66 (1.51)		22.82 (4.03)		27.03 (1.23)	
Time, baseline (vs. 6-month follow-up)	0.56 (0.40)	0.16	−0.54 (0.94)	0.58	1.11 (0.43)	0.01 *
Female sex (vs. male)	0.53 (0.58)	0.36	5.38 (30.05)	0.09	1.41 (0.65)	0.03 *
Age	0.07 (0.02)	0.001 **	0.01 (0.06)	0.83	0.006 (0.02)	0.74
College or above (vs. below college)	0.61 (0.63)	0.96	0.99 (1.58)	0.54	−0.27 (0.72)	0.71
Health insurance coverage (vs. no)	0.39 (0.75)	0.53	3.65 (3.08)	0.25	2.51 (0.75)	0.001 **

* *p* < 0.05; ** *p* < 0.01.

## Data Availability

The data presented in this study are available on request from the corresponding author.

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
