# Peer review of "Reducing Liver Cancer Risk through Dietary Change: Positive Results from a Community-Based Educational Initiative in Three Racial/Ethnic Groups"

_nutrients, 2022, doi:10.3390/nu14224878_

Round 1
Reviewer 1 Report
Liver cancer is the third leading cause of cancer-related death worldwide. Most patients are diagnosed at late stages with poor prognosis; thus, identification of modifiable risk factors for primary prevention of liver cancer is urgently needed. The well-established risk factors of liver cancer include chronic infection with hepatitis B virus (HBV) or hepatitis C virus (HCV), heavy alcohol consumption, metabolic diseases such as obesity and diabetes, and aflatoxin exposure. However, a large proportion of cancer cases worldwide cannot be explained by current known risk factors. Dietary factors have been suspected as important, but dietary aetiology of liver cancer remains poorly understood. In this article, the authors realize an interesting community-based education initiative for liver cancer prevention among the African, Asian, and Hispanic populations within the Greater Philadelphia and metropolitan New York City areas. They compared the scores of 11 specific food categories and the overall Mediterranean diet scores between baseline and the 6-month follow-up assessment within each of the three racial/ethnic groups. They observed that the Hispanic subgroup have significant change in dietary behaviors, with the Mediterranean dietary score increasing significantly from 30.000 at baseline survey to 31.187 after 6-month follow-up assessment. In the African Americans participants, the
consumption scores of fruits and poultry increased significantly, with a decresease in vegetables and red meat. At the end, In Asian Americans, the consumption of non-refined cereals, red meats, and dairy products decreased. Furthermore, the found that alcohol consumption decreased significantly onlyin Hispanics group. The significant changes above in Hispanic group is associated with the high rate of liver cancer in this group Furthermore, the authors report the suboptimal retention rate because of the significant effect of COVID-19 on education and employement. Furthermore, this rate is very low among the African American communities who they live in the hardest hit neighborhoods in the greater Philadelphia metropolitan and the greater NYC metropolitan area and in addtion, they had not a robust and sustainable academic-community. The paper is well written and structure, methods are explained and results are thoroughly discussed.
Major Comment
I think that the paper is dealing with issues that are of interest in different fields The only major issue I found missing in the paper is the lack of a validation procedure to to analyse potential correlation among sex, age, education and employment with eating habits within in the groups. These correlations should be investigated using the appropriate statistical techniques
Author Response
We appreciate the reviewer's comments and made revisions as suggested. Please see our point-to-point response attached.

Reviewer 2 Report
Line 97 provide NIH grant number(s)
Statistical analyses need to be corrected for multiple comparisons before any judgement can be made
Author Response
We appreciated the comments and suggestions from reviewer 2. Please see our point-to-point response attached.

Round 2
Reviewer 1 Report
The paper is very interesting
Author Response
Thank you for the comments.
Reviewer 2 Report
the authors did not address my concern about multiple comparisons. They are testing multiple hypotheses and therefore need to correct the level of significance. Can you send this out for additional statistical review? Now I am concerned they do not understand what multiple comparison correction is which is indicative of other potential statistical errors.Author Response
We sincerely appreciate the suggestion from Reviewer 2 and have provided a response along with the revised manuscript.
